# Implicit Generative Copulas

**Tim Janke, Mohamed Ghanmi, Florian Steinke**
Energy Information Networks and Systems
TU Darmstadt, Germany
`{tim.janke},{florian.steinke}@tu-darmstadt.de`

## Abstract

Copulas are a powerful tool for modeling multivariate distributions as they allow to separately estimate the univariate marginal distributions and the joint dependency structure. However, known parametric copulas offer limited flexibility especially in high dimensions, while commonly used non-parametric methods suffer from the curse of dimensionality. A popular remedy is to construct a tree-based hierarchy of conditional bivariate copulas. In this paper, we propose a flexible, yet conceptually simple alternative based on implicit generative neural networks. The key challenge is to ensure marginal uniformity of the estimated copula distribution. We achieve this by learning a multivariate latent distribution with unspecified marginals but the desired dependency structure. By applying the probability integral transform, we can then obtain samples from the high-dimensional copula distribution without relying on parametric assumptions or the need to find a suitable tree structure. Experiments on synthetic and real data from finance, physics, and image generation demonstrate the performance of this approach.

## 1 Introduction

Many approaches are available to reliably model univariate distributions of real-valued variables. One can use various parametric distributions or employ flexible, non-parametric methods, e.g., by using empirical distribution functions, kernel density estimation (KDE), or quantile regression [22]. Modeling distributions in higher dimensions is a much harder task. Parametric approaches using, e.g., Gaussian distributions, are common but inflexible. Many non-parametric approaches become infeasible in higher dimensions due to the curse of dimensionality [18].

Recent successes in modeling multivariate distributions have been achieved with deep neural network-based likelihood-free implicit generative models [28], such as Generative Adversarial Networks [13] (GANs) and Generative Moment Matching Networks (GMMNs) [10; 24].

Copulas are a tool to decouple the modeling of the univariate marginal distributions from modeling the high-dimensional joint dependency structure [19]. This allows to obtain high-quality marginal models with the mentioned univariate techniques. Moreover, the (conditional) marginals can be tailored to the problem at hand and many fields have developed sophisticated domain-specific univariate models for this task, e.g., for probabilistic forecasts in finance [4], weather [36], or energy [17]. These models can then be combined with a suitable copula structure to enable the simulation of multivariate quantities, e.g., [34; 29; 41]. In the machine learning literature copulas have been used to increase the flexibility of Bayesian networks [11], to model multi-agent coordination in reinforcement learning [43], or for image generation via the Vine Copula Autoencoder [40]. Another popular application of copulas is synthetic tabular data generation [33; 27]. Note that in many of these applications, the main goal is to sample from a multivariate distribution and the copula is used as a building block for a generative model.

35th Conference on Neural Information Processing Systems (NeurIPS 2021), virtual.

High-dimensional copula distributions are commonly modeled via parametric structures such as the Gaussian or the student-$t$ copula [8]. [25] propose to enhance the subclass of Archimedean copulas by learning the generator functions via deep neural networks. However, the family of Archimedean copulas is of limited use in higher dimension as it makes restrictive symmetry assumptions. A more flexible, semi-parametric state-of-the art approach is to use vine copulas which are built from trees of bivariate pair-copulas [1]. These pair-copulas can again be parametric, such as the Gumbel or Clayton copula, or non-parametric, e.g., by using bivariate kernel density estimation [32].

Directly applying implicit generative modeling to the task of estimating copula distributions is not straight-forward since copula distributions are required to have uniform marginals. Such an approach has been proposed by [23; 16], but without ensuring the uniformity property, at least not for finite sample sizes when the model is not expected to exactly fit the true copula distribution. Ensuring the marginal uniformity of the learned copula distribution is important since deviations from uniformity will result in unwanted alterations of the marginal distributions in the data space.

In this paper we show how to design and train implicit generative copula (IGC) models to match the dependency structure of given data. A key challenge is to ensure the marginal uniformity of the estimated distribution. We achieve this through first learning a latent distribution with unspecified marginals but the same dependency structure as the training data. We then obtain the desired copula model by applying the probability integral transform component-wise to the latent distribution. The IGC model and the data distribution are matched in copula space using the energy distance [39]. During training, the probability integral transform is approximated through a differentiable softrank layer which allows to use gradient-based methods.

**Our contributions**

- We propose the first universal, non-parametric model for estimating high-dimensional copula distributions with guaranteed uniformity of the marginal distributions.
- We show how flexible likelihood-free implicit generative models based on deep neural networks can be trained for this task through the use of a differentiable softrank layer.
- Compared with the state-of-the-art semi-parametric vine copula approach, we demonstrate similar or improved performance for different tasks.

We present the proposed copula model in Section 2. Model training is described in Section 3. Section 4 shows experimental results for synthetic and real data from finance, physics, and image generation. We conclude in Section 5.

## 2   Proposed IGC Model

We introduce our IGC model following the schematic shown in Figure 1. Figure 2 provides an exemplary application for a bivariate two-component Gaussian mixture data distribution.

**Copula Basics**   The vector-valued continuous random variable $\mathbf{X} \in \mathbb{R}^D$ represents the data source of our task. We denote its distribution by $P$ and the cumulative distribution function (cdf) of $\mathbf{X}$ by $F_{\mathbf{X}}(\mathbf{x})$. Moreover, let $F_{X_d}(x_d)$ be the cdf of the marginal distribution of $X_d$, the $d$-th component of $\mathbf{X}$, for $d = 1, \ldots, D$.

Each sample vector $\mathbf{x} \in \mathbb{R}^D$ can be mapped to a vector $\mathbf{u} \in [0, 1]^D$ by defining as $u_d$ the value of $x_d$ with respect to the corresponding marginal cdf of $X_d$, i.e., $u_d = F_{X_d}(x_d)$ for all $d = 1, \ldots, D$. This operation is called the probability integral transform (PIT). In the copula literature, values obtained by the PIT are called pseudo-observations [19]. The random variable $\mathbf{U}$ follows a so-called *copula distribution* $P^C$ by construction, i.e., the distribution is defined on the *unit space* $[0, 1]^D$ and its marginals are uniform. The joint cdf of $\mathbf{U}$ is typically called the *copula*, which we denote by $F_{\mathbf{U}}(\mathbf{u})$ here. Sklar's theorem states that such such a copula function exists for all random variables $\mathbf{X}$ and it holds that $F_{\mathbf{X}}(\mathbf{X}) = F_{\mathbf{U}}(F_{X_1}(X_1), \ldots, F_{X_d}(X_D))$, i.e., any multivariate distribution can be expressed in terms of its marginals and its copula [38].

**IGC Model**   In this work, we aim at defining a flexible, non-parametric model for copula distributions $Q_{\boldsymbol{\theta}}^C$ in high dimensions. Such a model can either be used to (approximately) represent the

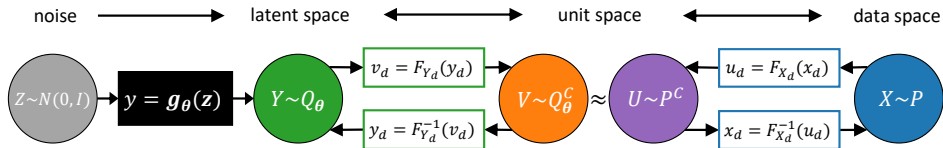

Figure 1: Illustration of the implicit generative copula (IGC) concept: We use a generative neural network $\mathbf{g}_{\boldsymbol{\theta}}$ with learnable parameters $\boldsymbol{\theta}$ to map samples from a multivariate standard Normal distribution to samples of the latent distribution $Q_{\boldsymbol{\theta}}$. The marginal cdfs $F_{Y_d}$ of this distribution are then used to generate samples of the copula distribution $Q_{\boldsymbol{\theta}}^C$. The model is trained to match the copula distribution of the real data $P^C$ using the energy distance.

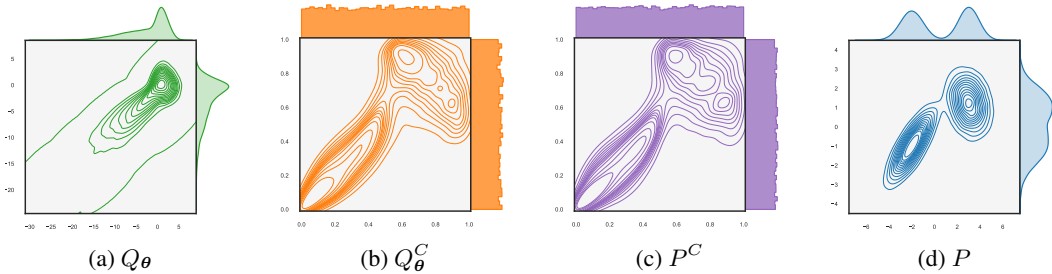

| (a) $Q_{\boldsymbol{\theta}}$ | (b) $Q_{\boldsymbol{\theta}}^C$ | (c) $P^C$ | (d) $P$ |

Figure 2: Contour plots and marginal densities of all data and model distributions mentioned in Figure 1 for the example of a two-dimensional mixture of Gaussians data distribution $P$. Contour plots are derived via KDE from a set of 10000 samples.

copula distribution $P^C$ of the unit space vectors $\mathbf{U}$ or the distribution $P$ of the original data vectors $\mathbf{X}$, in which case the $\mathbf{U}$ vectors have to be transformed component-wise with the inverse cdfs of the components $X_d$, i.e., $x_d = F_{X_d}^{-1}(u_d)$.

A flexible model class for learning high-dimensional probability distributions is the class of implicit generative models [28]. Here, latent random variables with a simple and known distribution are transformed via a parameterizable mapping, typically a deep neural network, to a complex distribution that approximates the distribution of the training data. A straight-forward application of this framework to model copula distributions, as done in [16] and [23], is difficult. The output of the generative model can be guaranteed to lie in $[0, 1]^D$ by applying an appropriate output layer, e.g., using a sigmoid function. However, guaranteeing the desired uniformity of the marginal distributions is not straight-forward. We have first experimented with additional training losses that penalize deviations from marginal uniformity. However, a simpler approach without any hyper-parameters for weighting additional cost terms and with a guarantee of marginal uniformity is the following two-step procedure.

In the first step, we model the distribution of the vector-valued latent random variable $\mathbf{Y} \in \mathbb{R}^D$. To this end, we start with random variables $\mathbf{Z} \in \mathbb{R}^K$ from a simple base distribution that we choose as zero mean, unit variance Gaussian here. The tuneable map $\mathbf{g}_{\boldsymbol{\theta}} : \mathbb{R}^K \to \mathbb{R}^D$ with parameters $\boldsymbol{\theta}$ is then used to transform the noise samples $\mathbf{z}$ to the latent samples $\mathbf{y}$ as

$$\mathbf{y} = \mathbf{g}_{\boldsymbol{\theta}}(\mathbf{z}). \tag{1}$$

We denote the resulting probability distribution of $\mathbf{Y}$ by $Q_{\boldsymbol{\theta}}$ and the marginal cdfs by $F_{Y_d}(y_d)$.

In a second step, we transform the latent variables $\mathbf{Y}$ into unit space vectors $\mathbf{V}$. To this end, we define component-wise for $d = 1, \ldots, D$

$$v_d = F_{Y_d}(y_d). \tag{2}$$

This transformation step is analogous to going form $\mathbf{X}$ to $\mathbf{U}$. It guarantees both that the sample values $v_d$ from the model lie in the unit interval $[0, 1]$ and that they are uniformly distributed, regardless of the distribution $Q_{\boldsymbol{\theta}}$. We refer to the distribution of $\mathbf{V}$ as the model copula distribution and denote it by $Q_{\boldsymbol{\theta}}^C$.

**Interpretation** Note that the distributions $Q_\theta$ and $P$ might be very different. More precisely, the mapping (1) can result in arbitrary marginal distributions but an optimal model of the distribution $P$ would have

$$P^C = Q_\theta^C. \tag{3}$$

Given a sufficiently rich function class for $\mathbf{g}_\theta$ any distribution of $\mathbf{Y}$ can be modeled with small error [26]. This statement naturally extends to universal approximation properties for the dependency structure of $\mathbf{Y}$, i.e., the unit space random vectors $\mathbf{V}$. The approximation of the data copula distribution $P^C$ with our model copula distribution $Q_\theta^C$ will thus be arbitrarily close if the generator function $\mathbf{g}_\theta$ is flexible enough and if the optimal $\theta$ is selected.

## 3 Training Procedure

**Problem statement** We aim at empirically estimating an IGC model $Q_\theta^C$ from data, i.e., we want to determine the optimal parameter vector $\theta$ given a set of $N$ training samples $\mathbf{x}^1, \ldots, \mathbf{x}^N$ from $\mathbf{X}$. The underlying target is to optimally match the copula model $Q_\theta^C$ to the (typically unknown) copula distribution $P^C$ from which the samples were generated.

**Parameter estimation** We first transform the given data samples into unit space vectors in $[0,1]^D$, before starting the actual training procedure. Since we typically do not know the exact marginal cdfs $F_{X_d}$ of the components $X_d$, we estimate them from the given data. To this end, we use the empirical cdfs separately for each component and define for $n = 1, \ldots, N$ and $d = 1, \ldots, D$,

$$u_d^n = \frac{1}{N} \sum_{i=1}^N \mathbf{1}[x_d^i \le x_d^n]. \tag{4}$$

Here, $\mathbf{1}[A]$ is the indicator function of event $A$, i.e., it is 1 iff $A$ is true. Note that this is an unbiased and consistent estimator for the true cdf [42].

We then aim at matching the copula distribution $Q_\theta^C$ to the unit space vector samples $\mathbf{u}^1, \ldots, \mathbf{u}^N$. We base the inference of our likelihood-free model on the energy distance $ED^2$ [39] between distributions $P^C$ and $Q_\theta^C$,

$$ED^2(P^C, Q_\theta^C) = 2\mathbb{E}\|\mathbf{U} - \mathbf{V}\| - \mathbb{E}\|\mathbf{U} - \mathbf{U}'\| - \mathbb{E}\|\mathbf{V} - \mathbf{V}'\|. \tag{5}$$

Here, $\mathbf{U}, \mathbf{U}'$ and $\mathbf{V}, \mathbf{V}'$ are independent copies of a random vector with distribution $P^C$ and $Q_\theta^C$, respectively. It holds that $ED^2(P^C, Q_\theta^C) = 0$ if and only if $P^C = Q_\theta^C$ [39].

In practice, we use a sample-based approximation of the energy distance. We draw $M$ samples $\mathbf{v}^1(\theta), \ldots, \mathbf{v}^M(\theta)$ from $Q_\theta^C$ and minimize the loss function

$$\mathcal{L}_{ED^2}(\theta) = \frac{1}{NM} \sum_{n=1}^N \sum_{m=1}^M \|\mathbf{u}^n - \mathbf{v}^m(\theta)\|_2 - \frac{1}{2M(M-1)} \sum_{m=1}^M \sum_{m'=1}^M \|\mathbf{v}^m(\theta) - \mathbf{v}^{m'}(\theta)\|_2. \tag{6}$$

Since $P^C$ is independent of $\theta$, the second term of (5) does not need to be included. Note that the energy distance is an instance of the more general maximum mean discrepancy (MMD) measure [15; 37]. We also experimented with MMD losses based on the Gaussian kernel but found the energy distance to be more robust as it does not require to tune the kernel bandwidth.

To generate samples from our copula model $Q_\theta^C$, we first draw samples $\mathbf{y}^1, \ldots, \mathbf{y}^M$ from the implicit generative model $Q_\theta$. To obtain vectors in unit space, we then again require the cdfs $F_{Y_d}(y_d)$ of the marginal distributions of the components $Y_d$ of $\mathbf{Y}$, $d = 1, \ldots, D$. However, these functions are not known during model training and will likely change after each gradient step. Hence, we again use the empirical distribution functions to define for $m = 1, \ldots, M$ and $d = 1, \ldots, D$,

$$v_d^m(\theta) = \frac{1}{M} \sum_{j=1}^M \mathbf{1}[y_d^j(\theta) \le y_d^m(\theta)]. \tag{7}$$

This provides us with an unbiased estimate of the current cdf during training. However, the indicator operation in (7) is not differentiable and hence will not allow the gradients to flow through this

operation. Therefore, during training, we replace (7) with a *softrank* layer based on a scaled sigmoid function

$$v_d^m(\boldsymbol{\theta}) = \frac{1}{M}\left(0.5 + \sum_{j=1}^{M} \frac{1}{1 + \exp(\alpha(y_d^m(\boldsymbol{\theta}) - y_d^j(\boldsymbol{\theta})))}\right), \tag{8}$$

where $\alpha$ is a scaling constant [35]. For sufficiently large $\alpha$, (8) provides a close approximation to the empirical marginal cdfs at the current training step. A larger $M$ will result in a finer approximation of the true marginal cdfs but comes with the increased cost for computing the ranks which requires $\mathcal{O}(M^2)$ operations for all samples.

**Sampling from the trained model**   Once we have completed training and thus have fixed $\boldsymbol{\theta}$, we have also fixed the component-wise cdfs $F_{Y_d}(y_d)$. Now, the operation to estimate $F_{Y_d}(y_d)$ does not need to be differentiable anymore and we can chose any available method to estimate the univariate marginal distributions based on samples from $Q_{\boldsymbol{\theta}}$. We choose to simply draw a very large set of samples from $Q_{\boldsymbol{\theta}}$ and then use the empirical marginal cdfs, i.e., for each sampled value $y_d^t$ we store the corresponding value $u_d^t$ according to (4), $t = 1, \ldots, T$. Given a new realization of $Y_d$, we obtain its approximate cdf value $\hat{F}_{Y_d}(y_d)$ by interpolation of the stored values. Other more compact approximations like univariate kernel density estimation would also be feasible.

In sum, we obtain a sample $\mathbf{v}^i$ from $Q_{\boldsymbol{\theta}}^C$ at test time by first sampling a realization $\mathbf{z}^i$ from $N(\mathbf{0}, \mathbf{I})$, transform it via $\mathbf{y}^i = \mathbf{g}_{\boldsymbol{\theta}}(\mathbf{z}^i)$, and then apply the component-wise PIT approximation to obtain $\mathbf{v}^i = [\hat{F}_{Y_1}(y_1^i), \ldots, \hat{F}_{Y_d}(y_D^i)]$. Given estimates for the component-wise marginal cdfs of the data $\hat{F}_{X_1}, \ldots, \hat{F}_{X_d}$ we can derive a sample $\mathbf{s}^i$ in data space via $\mathbf{s}^i = [\hat{F}_{X_1}^{-1}(v_1^i), \ldots, \hat{F}_{X_d}^{-1}(v_D^i)]$.

## 4   Experiments

In the following we empirically demonstrate the capabilities of IGC models on a series of experiments on synthetic and real data with increasing complexity. Additional experiments can be found Appendix B.

**Implementation**   For all experiments except the image generation task, we use a fully connected neural network with two layers, 100 units per layer, ReLU activation functions, and train for 500 epochs. For the image generation experiment we use a three layer, fully connected neural network with 200 neurons in each layer, and train for 100 epochs. In all cases we train with a batch size of $N_{batch} = 100$ and generate $M = 200$ samples from the model per batch. The number of noise distributions is set as $K = 3D$ and $T = 10^6$. We use the Adam optimizer [20] with default parameters.

All experiments besides the training of the autoencoder models were carried out on a desktop PC with a Intel Core i7-7700 3.60Ghz CPU and 8GB RAM. For the training of the autoencoders we used Google Colab [14]. Training times for all experiments are in the range of a few minutes except for the image generation task. Details are provided in Appendix B. We use tensorflow with the Keras API for the neural networks [2; 7]. For copula modeling, we use pyvinecopulib [31] in Python and the R package kdecopula [30]. Our code is available from `https://github.com/TimCJanke/igc`.

**Evaluation**   We evaluate the models by comparing the distance between the learned and the true copula. More specifically, we use the integrated squared error (ISE) in unit space, i.e.,

$$ISE = \int_{[0,1]^D} (F_{\mathbf{U}}(\mathbf{w}) - F_{\mathbf{V}}(\mathbf{w}))^2 d\mathbf{w} \approx \frac{1}{N}\sum_{n=1}^{N}(F_{\mathbf{U}}(\mathbf{u}^n) - F_{\mathbf{V}}(\mathbf{u}^n))^2, \tag{9}$$

where $F_{\mathbf{U}}$ is the joint cdf of $\mathbf{U}$, the true copula, and $F_{\mathbf{V}}$ the joint cdf of the learned model. Since analytical integration is not possible, we approximate the integral with an empirical sum and use the data vectors $\mathbf{u}^1, \ldots, \mathbf{u}^N$ for this purpose.

The copula function $F_{\mathbf{U}}$ is not known for real data. Instead, we use the empirical cdf of the unit space data vectors in (9), i.e., for $\mathbf{w} \in [0,1]^D$ we set

$$\hat{F}_{\mathbf{U}}(\mathbf{w}) = \frac{1}{N}\sum_{n=1}^{N}\prod_{d=1}^{D}\mathbf{1}[u_d^n \le w_d]. \tag{10}$$

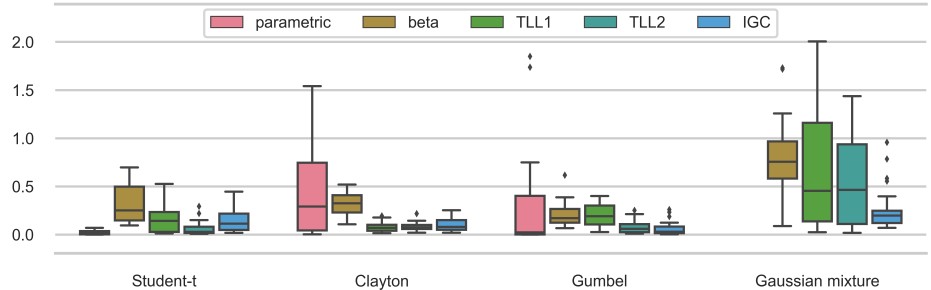

Figure 3: Box plot for the integrated squared error (lower is better) for 25 runs of fitting bivariate copulas to training data from a Student-t, Gumbel, Clayton, and Gaussian mixture copula.

Due to the curse of dimensionality this approximation of the joint cdf will have a coarse structure in high dimensions. This is why we only use the values at the data vectors $\mathbf{u}^1, ..., \mathbf{u}^N$ in (9).

For parametric copula models the distribution function $F_{\mathbf{V}}$ is analytic. For the non-parametric models, including IGC, we again resort to approximation. We draw $10^5$ samples from the copula distribution model and use their empirical cdf in unit space in (9). We found that this procedure provided reliable and stable estimates for the underlying copula in dimensions $D \leq 5$.

## 4.1 Synthetic data

**Learning bivariate parametric copulas**   We first show that IGC models are able to emulate bivariate parametric copulas. We compare our IGC approach to two non-parametric copula density estimation techniques, kernel density estimation with beta kernels (BETA) [6] and the transformation local likelihood estimator (TLL) [12]. These are state-of-the art non-parametric estimators for pair-copulas [30]. The compact support of the beta distribution on $[0, 1]$ is convenient for copula density estimation. The idea of the TLL approach is to first transform the data using the inverse normal CDF such that the data is supported on $\mathbb{R}^2$. Then the density is estimated via local regression using linear (TLL1) or cubic polynomials (TLL2) on a fixed grid of points. Finally the estimated density is transformed back to unit space. Additionally we report the performance of a parametric model. In this approach a copula family is selected from a set of copulas based on the Bayesian information criterion (BIC). This set comprises the Independence, Gaussian, Student-t, Clayton, Gumbel, Frank, Joe, BB1, and BB7 copula. Parameters are then estimated via maximum likelihood.

We run experiments for data sets generated from a Student-t, a Gumbel, and a Clayton copula as well as the copula resulting from a two-component Gaussian mixture distribution as shown in Figure 2. For each one of these, we sampled a random parameter set and then generated 1000 training samples from the resulting model. We repeated each experiment 25 times. Appendix B contains the details on the considered parameter ranges and the exact sampling procedure.

Figure 2 shows the data and the model with intermediate steps for one instance of the Gaussian mixture case. The fitted copula matches the data distribution in unit space very well. Figure 3 presents aggregated ISE results for the different setups. The IGC model shows comparable performance scores to TLL1 and TLL2 for the Student-t, Clayton, and Gumbel copulas, and better performance than the BETA approach. For the more complex Gaussian mixture test case, IGC shows superior performance and a much lower variance than all baseline methods. Notably, the parametric approach with BIC-based model selection results in large variance in accuracy for the Clayton and Gumbel data, most likely because the wrong copula family is selected in some cases.

**Learning multivariate vine copulas**   We now turn to the problem of estimating copulas in dimensions $D > 2$. To this end, we conduct a similar experiment as before using 5-dimensional vine copulas as data generating distribution. For each of 25 repetitions, we first sample a random tree structure using pyvinecopulib's `RVineStructure.Simulate()` method. Then, we randomly assign parametric pair-copulas with random parameters to each edge in the tree. The considered families of bivariate copulas are Independence, Gaussian, Student-t, Clayton, Gumbel, Frank, Joe, BB1, and BB7. Further details are available in Appendix B. We generate 5000 samples from the resulting vine copula

model and use these to train our IGC model as well as the benchmark models. As benchmarks, we use a vine copula model with $TLL2$ pair-copulas only and a vine copula model which can select all parametric copula families named above as well as the $TLL2$. Additionally, we estimate a parametric Gaussian copula. The parameters are estimated via maximum likelihood and the copula families are selected using the BIC. The selection of the vine structure is based on the Dissmann algorithm [9]. We evaluate all models with the ISE at 10000 data points sampled from the true model.

The results of the simulation study are presented in Figure 4. The IGC model has the lowest mean ISE (0.0697) followed by the TLL2-Vine model (0.2093), while the latter has a lower median ISE (0.0339 vs. 0.045). This is the result of some large outliers of the ISE for the vine models that do not occur for the IGC model. These are most likely caused by a poorly selected vine structure. Interestingly, the results specifically imply that the data is on average better modeled by the IGC model than by the vine approach although this model class entails the true data generating model. However, recovering the true model does not seem to be an easy task.

## 4.2 Real data

**Exchange rates**   Copulas are a popular tool in financial risk management as they can be used to estimate the distribution of the multivariate returns over different assets [34] under the assumption of a stationary copula. We consider a data set of size $N = 5844$ that contains 15 years of daily exchange rates between the US-Dollar and the Canadian Dollar, the Euro, the British Pound, the Swiss Franc, and the Japanese Yen. The data was obtained from the *R* package *qrm_data*. We preprocess the data to filter out the effects of temporal dependencies. To this end, we fit an AR(1)-GARCH(1,1) [4] process with Student-t innovations to the time series of the daily returns. We then obtain the standardized residuals from the AR-GARCH models and transform these observations to the unit space using the empirical cdfs. Figure 5a shows a kernel density estimate for two selected dimensions of the resulting data set in unit space, namely the US-Dollar/Euro and US-Dollar/Pound exchange rates. While being non-trivial and multi-modal, the distribution is strongly concentrated along the diagonal.

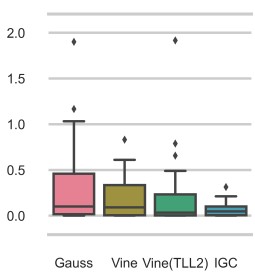

Figure 4: Box plot of the ISE results for 25 runs of fitting data from a 5D vine copula

We then estimate a Gaussian copula, a vine copula, a vine copula with only TLL2 pair-copulas, a GMMN as proposed by [16], and our IGC model for this data. The GMMN model has the same loss function, architecture, and hyper parameters as the IGC model but uses a sigmoid activation in the final layer. To ensure a test data set of appropriate size, we use a 5-fold cross-validation scheme where 20% of the data is used for training and 80% for testing. For the IGC and the GMMN model we additionally use five different random initializations per fold for the neural network weights, i.e., we report means and standard deviations from 25 values for the IGC and the GMMN model and 5 values for the other methods. The results are presented in Table 1. The IGC model clearly outperforms the Gaussian baseline and also the GMMN model. However, the vine copula models show lower average ISEs. This might be due to the symmetric nature of the data which can be approximated well by Student-t and TLL pair-copulas.

**MAGIC Gamma Telescopes data**   In order to test our approach on a more complex dependency structure, we consider the MAGIC (Major Atmospheric Gamma-ray Imaging Cherenkov) Telescopes data set available from the UCI repository (`https://archive.ics.uci.edu/ml/datasets/MAGIC+Gamma+Telescope`). This data was also used for benchmarking non-parametric copula estimation techniques in [32]. We only consider the observations classified as *gamma* and the 5 variables *fLength, Width, fConc, fM3Long, fM3Trans*. The size of the data set is $N = 12332$. We use the empirical cdfs to transform the

Table 1: ISE mean and standard deviation for exchange rate and magic data from 5 fold CV and 5 random NN weight initializations per fold (lower is better).

|  | exchange rates | magic |
|---|---|---|
| Gauss | $0.9196 \pm 0.1024$ | $0.2415 \pm 0.0167$ |
| Vine | $0.2791 \pm 0.0290$ | $0.0588 \pm 0.0055$ |
| Vine-TLL2 | $\mathbf{0.2457} \pm 0.0383$ | $0.0588 \pm 0.0055$ |
| GMMN | $0.5791 \pm 0.2638$ | $0.08065 \pm 0.0286$ |
| IGC (ours) | $0.3829 \pm 0.1385$ | $\mathbf{0.0345} \pm 0.0105$ |

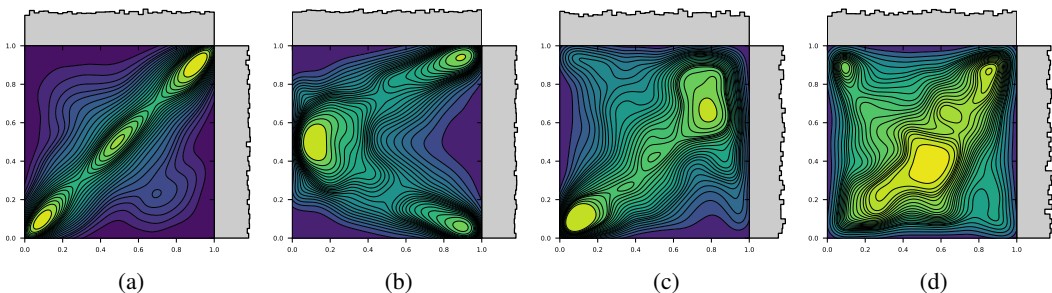

Figure 5: Exemplary bivariate KDE estimates of the copula densities from the real data sets: (a) Exchange rates EUR/USD-GBP/USD, (b) MAGIC fLength-fM3Trans, (c) and (d): FashionMNIST Autoencoder latent space dimensions 1-2 and 2-3

observations to the unit space. Figure 5b showcases the dependency structure of the data for two selected data dimensions. The observed structures are highly asymmetric. We again fit the data with a Gaussian copula, a vine copula with all available pair-copulas as above, a vine copula with TLL2 pair-copulas, and a GMMN model with a sigmoid output layer [16] as benchmarks and use the same 5-fold CV strategy as before, i.e., in each fold we use 20% as training data, 80% as test data, and report the average ISE. The results are presented in Table 1. The IGC model achieves the lowest average ISE. Again the GMMN model shows a substantially larger mean ISE with a much larger standard deviation. As only TLL2 pair-copulas were selected by the BIC criterion, the scores for both vine models are the same. The Gaussian copula is clearly not suited for such complex data as the average ISE is 5 times larger than for the other approaches.

**Copula Autoencoders**    [40] introduced the Vine Copula Autoencoder, a generative model which uses vine copulas for ex-post density estimation of the latent space of a trained autoencoder. In a first step, an autoencoder is trained on a data set to learn a low dimensional representation of the data. After training, the encoder part of the network is used to map the training data to the autoencoder's latent space representation. Next, the uni-variate marginal distributions are estimated, e.g., by using the empirical cdfs or kernel density estimation, and the compressed data is mapped to unit space. After fitting a copula model to the observations in unit space, one can sample from the fitted copula model, apply the inverse marginal cdfs, and map the simulated data back to the image space using the decoder network of the autoencoder in order to generate new data samples.

In the following, we present results for the FashionMNIST [44] data set. We train a convolutional autoencoder on the entire training data of 60000 samples with a latent space of dimension 25. Details on the architecture are found in Appendix B. After training the autoencoder, we estimate the empirical cdfs of the compressed data using the training data. See Figures 5c and 5d for exemplary visualizations of the resulting bivariate data densities in unit space. Notably, these distributions are more complex than the ones from the previous experiments. We fit this data with a Gaussian copula, a vine copula with TLL2 pair-copulas, a GMMN with sigmoid output layer

Table 2: MMD scores for the FashionMNIST test set (lower is better)

|        | image       | latent      | copula      |
|--------|-------------|-------------|-------------|
| Indep  | 0.01912     | 0.00722     | 0.00373     |
| Gauss  | 0.00619     | 0.00209     | 0.00087     |
| Vine   | 0.00674     | 0.00131     | 0.00079     |
| GMMN   | **0.00392** | 0.00341     | **0.00073** |
| IGC    | **0.00426** | **0.00114** | **0.00069** |
| VAE    | 0.01316     |             |             |

[16], and an IGC model. We also test an independence copula, i.e., we assume independence over the latent space. Additionally we report results for a standard variational autoencoder (VAE) [21] with the same architecture.

Exemplary samples from all models and the test set are presented in Figure 6. Sampling with the independence copula, i.e., assuming no dependency structure, leads to images with many artifacts. The Gaussian copula produces better images, but still produces some artifacts. Samples from the vine copula or the IGC model are comparable in quality. They show smaller details and few implausible artifacts. The images generated by the VAE are blurry and show much less variation in pixel intensity than the test data.

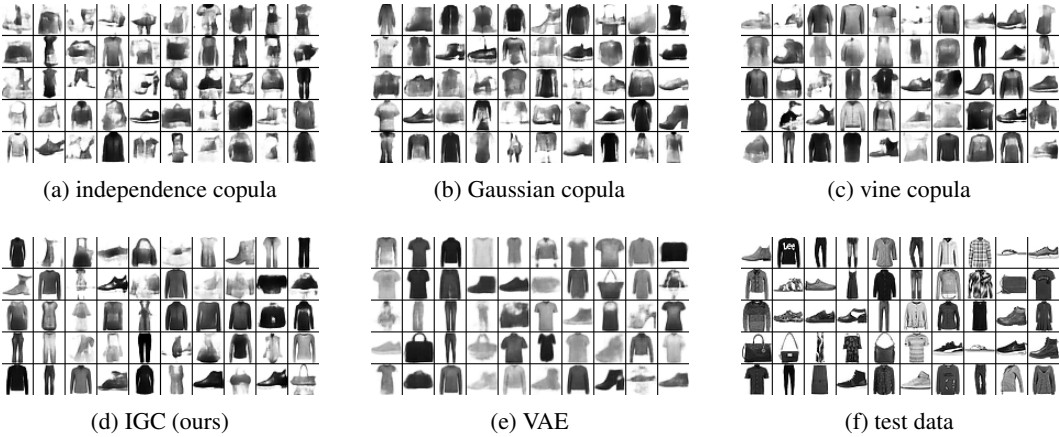

(a) independence copula        (b) Gaussian copula        (c) vine copula

(d) IGC (ours)               (e) VAE               (f) test data

Figure 6: Generated image samples from different copula models on top of a pre-trained autoencoder as well as from a standard variational autoencoder. The latent space has 25 dimensions.

In 25 dimensions it is not feasible anymore to compute and compare empirical cdfs as required for ISE scoring. We thus resort to the MMD [15] for the numerical evaluation since it is commonly used as a simple and robust measure for evaluating image generation models [46]. We generate 10000 images from each model and compare those to the 10000 test set images by computing the MMD with a Gaussian kernel. We use the bandwidths $1.5$, $350$, and $8$ for the latent unit space, the latent data space, and the image space, respectively. The bandwidths were selected based on the median heuristic proposed in [15].

The results are found in Table 2. We provide p-values for these results using the test proposed by [5] in Appendix B. The IGC and GMMN models perform better than all other models for the latent unit space distribution as well as the image space. Interestingly the GMMN model shows a relatively large MMD value for the latent data space. This could be caused by deviations from marginal uniformity of the learned copula distribution which alter the marginal distributions in the data space.

## 5 Conclusion

We have proposed the first fully non-parametric model framework for copulas in higher dimensions that guarantees uniformity of the marginal distributions. The proposed IGC approach, which is based on an implicit generative step and a differentiable ranking transformation, is structurally simple. Yet, if the generator class is sufficiently complex, any copula dependency structure can be modeled. The model can be well implemented with standard deep learning frameworks. For various data sets we have shown a modeling performance on par or above other state-of-the-art approaches, especially, the vine copula approach.

IGC models should be further investigated in various ways. First, it would be straight-forward to condition the model on external factors, either by including such values into the input of the generator network or by reparameterization of the noise distributions. This would allow to describe context-dependent changes of the dependency structure. Second, many more applications for copulas should be examined, e.g., synthetic tabular data generation [45]. Third, the ranking operation could probably be sped up from $\mathcal{O}(M^2)$ to $\mathcal{O}(M \log M)$ using ideas from [3]. Finally, the use of adversarial training schemes could also be investigated.

## Acknowledgments and Disclosure of Funding

This work has been performed in the context of the LOEWE center emergenCITY.

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
