# Appendix

## A   Algorithm

The algorithm for training an IGC model is described in Algorithm 1.

---

**Algorithm 1:** Training algorithm

---

**Input**   : data $\{\mathbf{u}_n\}_{n=1}^N$, initial model parameters $\boldsymbol{\theta}_0$, initial learning rate $\eta$, number of batches $B$, number of random noise samples $M$, number of epochs $N_{epochs}$
**Output** : Model parameters $\boldsymbol{\theta}^*$
Partition data into $B$ mini batches $\{\mathbf{U}_b\}_{b=1}^B$
For each batch $b$, generate a set of $M$ random noise samples $\mathbf{Z}_b = [\mathbf{z}_b^1, ..., \mathbf{z}_b^M]$
**for** $N_{epochs}$ **do**
    **for** $b = 1, ..., B$ **do**
        **for** $m = 1, ..., M$ **do**
           | Get $\mathbf{y}_b^m \leftarrow g(\mathbf{z}_b^m, \boldsymbol{\theta})$
        **end**
        Map model output to unit space $\mathbf{V}_b \leftarrow Softrank(\mathbf{Y}_b)$
        Compute gradient of loss $\nabla_{\boldsymbol{\theta}} \mathcal{L}(\mathbf{U}_b, \mathbf{V}_b)$ over batch
        Update learning rate $\eta$ (e.g. using ADAM)
        Update model parameters $\boldsymbol{\theta} \leftarrow \boldsymbol{\theta} - \eta \nabla_{\boldsymbol{\theta}} \mathcal{L}$
    **end**
**end**

---

## B   Experiments

### B.1   Additional experiments on toy data sets

We provide additional experiments on three toy data sets commonly used in the literature on deep generative models, the "Swiss Roll", the "Grid of Gaussians", and the "Ring of Gaussians". We test four copula based models: a Gaussian copula, a TLL2 copula, an GMMN with sigmoid output layer [16], and our IGC model. For these models we use a linear interpolation of the ECDF of the training set as models for the marginals of the data distribution. We additionally report results for two implicit generative models that directly model the data distribution, a GMMN [24; 10] and a GAN [13]. Like the IGC model, the GMMN based models are trained by minimizing the energy distance. All neural network models use two layers, 100 neurons per layer, and are trained for 500 epochs. IGC and GMMN models are trained using the Adam optimizer with standard values. For the GAN we use a lower learning rate of $lr = 0.0002$ and a lower momentum $\beta_1 = 0.5$ as the model did not converge with the standard settings. We evaluate the models using the average negative log-likelihood of a test set based on kernel density estimates. We repeat each experiment 10 times with different training and test sets of size 5000 and random initializations for the neural networks.

Table 3 presents the results. The GAN achieves the best result for the Swiss Roll data. However, the scores for the GAN show a much higher standard deviation than all other methods. The IGC model has the second lowest score overall and the lowest score of all copula based methods. For the ring of Gaussians, the TLL2 copula shows the lowest NLL, closely followed by the IGC and GMMN copula. Here the GAN shows the worst performance of all models. The Grid of Gaussians is a trivial test case for the copula based models as it is sufficient to sample from the marginal distributions with an independence copula. Both the GMMN and the GAN show substantially worse NLL values and fail at properly approximating the true data distribution as can be seen from the bottom row of Figure 7.

### B.2   Learning bivariate copulas

For the Student-t, Gumbel, and Clayton copulas we sample the parameters and rotations uniformly from the ranges given in Table 4.

Samples form the Gaussian mixture copula are generated by sampling $N$ times from one of the two components with equal probability and then transforming all samples to the unit space via the

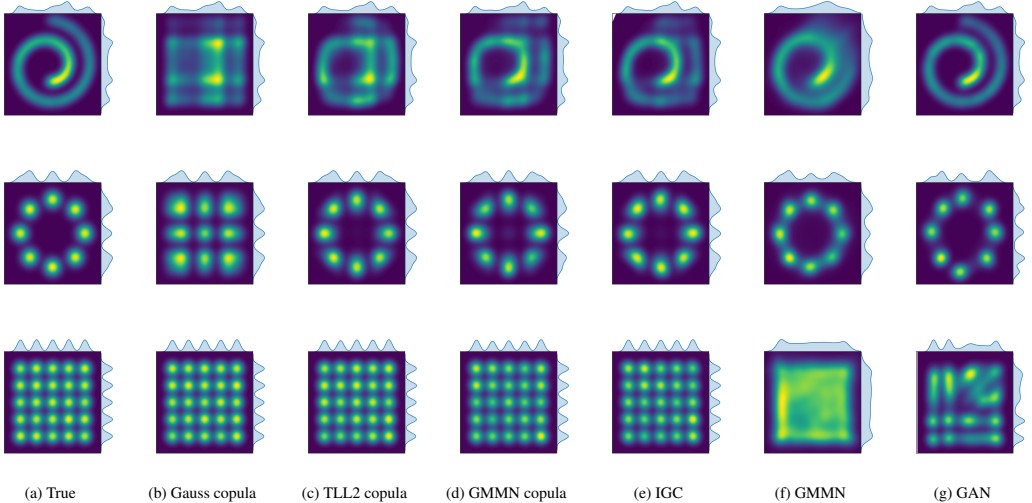

| | (a) True | (b) Gauss copula | (c) TLL2 copula | (d) GMMN copula | (e) IGC | (f) GMMN | (g) GAN |

Figure 7: True and exemplary learned densities for the Swiss Roll (top), Ring of Gaussians (middle), and Grid of Gaussians (bottom).

Table 3: Means and standard deviations of the average negative log-likelihood (lower is better) from 10 repetitions with random parameter initializations and training and test sets.

| | swiss roll | ring | grid |
|---|---|---|---|
| Gaussian copula | 6.10 (0.01) | 5.72 (0.02) | **4.47 (0.01)** |
| TLL2 copula | 5.61 (0.01) | **5.11 (0.01)** | **4.47 (0.01)** |
| GMMN copula | 5.45 (0.09) | 5.24 (0.01) | 4.48 (0.01) |
| IGC | 5.26 (0.08) | 5.19 (0.03) | **4.47 (0.01)** |
| GMMN | 5.76 (0.09) | 5.40 (0.05) | 6.27 (0.05) |
| GAN | **4.82 (0.40)** | 7.14 (0.52) | 5.95 (0.26) |

component-wise PIT. Samples for the component $j \in \{1, 2\}$ are drawn from the two-dimensional Gaussian distribution $\mathcal{N}(\boldsymbol{\mu}_j, \alpha_j \boldsymbol{\Sigma}_j)$, where $\boldsymbol{\mu}_j$ is sampled from $\mathcal{U}(-5, 5)^2$, $\sigma_{12,j}$ is sampled from $\mathcal{U}(-0.95, 0.95)$, $\alpha_j$ is sampled from $\mathcal{U}(0.8, 1.2)$, and $\boldsymbol{\Sigma}_j = \left( \begin{smallmatrix} 1 & \sigma_{12,j} \\ \sigma_{12,j} & 1 \end{smallmatrix} \right)$.

### B.3 Learning multivariate vine copulas

The following table presents the parameter ranges of the pair-copulas for the vine copula experiments. We used uniform sampling for selecting the pair-copula family as well as the parameters and rotations.

Table 4: Parameter ranges for pair-copulas for learning vine copulas

| | $\theta_1$ | $\theta_2$ | rotation |
|---|---|---|---|
| Independence | - | - | - |
| Gaussian | $\pm[0.5, 0.95]$ | - | $0°$ |
| Student-t | $\pm[0.5, 0.95]$ | $[2, 10]$ | $0°$ |
| Clayton | $[2, 10]$ | - | $\{0°, 90°, 180°, 270°\}$ |
| Gumbel | $[2, 10]$ | - | $\{0°, 90°, 180°, 270°\}$ |
| Frank | $[10, 25]$ | - | $0°$ |
| Joe | $[2, 10]$ | - | $\{0°, 90°, 180°, 270°\}$ |
| BB1 | $[1, 5]$ | $[1, 5]$ | $\{0°, 90°, 180°, 270°\}$ |
| BB7 | $[1, 6]$ | $[2, 20]$ | $\{0°, 90°, 180°, 270°\}$ |

## B.4 Training times

Table 5 shows the average training times for the IGC model and a vine copula model with TLL2 pair-copulas for the different data sets. Note that the IGC model for the FashionMNIST experiment is a three-layer neural network with 200 neurons per layer which is trained for 100 epochs while in all other cases a two-layer network with 100 neurons per layer was trained for 500 epochs. Timings are for a desktop PC with a Intel Core i7-7700 3.60Ghz CPU and 8GB RAM.

Table 5: Average training times for IGC and vine copula (TLL2)

|  | IGC | vine copula (TLL2) | $D$ | $N_{train}$ |
|---|---|---|---|---|
| Learning bivariate copulas | 49s | <1s | 2 | 1000 |
| Learning vine copulas | 131s | 5s | 5 | 5000 |
| Exchange rates | 33s | 4s | 5 | 1169 |
| Magic | 61s | 4s | 5 | 2466 |
| FashionMNIST AE | 1472s | 1743s | 25 | 60000 |

## B.5 Autoencoder architecture

We used the following architecture for the autoencoder and the VAE:

- Encoder:

$$x \in \mathbb{R}^{32 \times 32 \times 1} \to Conv(32, 4, 2) \to BN \to ReLU$$
$$\to Conv(64, 4, 2) \to BN \to ReLU$$
$$\to Conv(128, 4, 2) \to BN \to ReLU$$
$$\to FC(256) \to ReLU$$
$$\to FC(25) \to z \in \mathbb{R}^{25} \;\; (AE)$$
$$\to FC(50) \to \mathcal{N}(\boldsymbol{\mu}, \boldsymbol{\sigma}\mathbf{I}) \to z \in \mathbb{R}^{25} \;\; (VAE)$$

- Decoder:

$$z \in \mathbb{R}^{25} \to FC(4096) \to ReLU \to Reshape((4, 4, 256))$$
$$\to ConvT(128, 4, 2) \to BN \to ReLU$$
$$\to ConvT(64, 4, 2) \to BN \to ReLU$$
$$\to ConvT(32, 4, 2) \to BN \to ReLU$$
$$\to ConvT(1, 1, 1) \to Sigmoid \to y \in \mathbb{R}^{32x32x1}$$

$Conv(a, b, c)$ denotes a convolutional layer with $a$ filters, a kernel of height and width $b$, and a stride with height and width $c$. $ConvT(a, b, c)$ denotes a deconvolutional layer. $FC(a)$ denotes a fully connected layer with $a$ neurons. $Sigmoid$ and $ReLU$ denote the sigmoid and ReLU activation functions and $BN$ denotes batch normalization layers. We use a padding of 2 to achieve an image resolution of $32 \times 32$ pixels and normalize the inputs to the range $[0, 1]$. The models are trained with the binary cross entropy as reconstruction loss for 100 epochs using the Adam optimizer with default parameters.

## B.6 Significance of FashionMNIST results

In the following tables we present the p-values from the test proposed in [5] for the experiments on the FashionMNIST data. Values close to one/zero in the row indicate significantly better/worse performance compared to model in the column.

Table 6: p-values for copula space samples

|       | IGC    | GMMN   | Vine   | Gauss  | Indep  |
|-------|--------|--------|--------|--------|--------|
| IGC   |        | 0.5561 | 0.9871 | 0.9990 | 1.0000 |
| GMMN  | 0.4439 |        | 0.8645 | 0.9689 | 1.0000 |
| Vine  | 0.0129 | 0.1355 |        | 0.9510 | 1.0000 |
| Gauss | 0.0010 | 0.0311 | 0.0490 |        | 1.0000 |
| Indep | 0.0000 | 0.0000 | 0.0000 | 0.0000 |        |

Table 7: p-values the latent space samples

|       | IGC    | GMMN   | Vine   | Gauss  | Indep  |
|-------|--------|--------|--------|--------|--------|
| IGC   |        | 1.0000 | 0.9812 | 1.0000 | 1.0000 |
| GMMN  | 0.0000 |        | 0.0000 | 0.0000 | 1.0000 |
| Vine  | 0.0188 | 1.0000 |        | 1.0000 | 1.0000 |
| Gauss | 0.0000 | 1.0000 | 0.0000 |        | 1.0000 |
| Indep | 0.0000 | 0.0000 | 0.0000 | 0.0000 |        |

Table 8: p-values image space samples

|       | IGC    | GMMN   | Vine   | Gauss  | Indep  | VAE    |
|-------|--------|--------|--------|--------|--------|--------|
| IGC   |        | 0.1131 | 1.0000 | 1.0000 | 1.0000 | 1.0000 |
| GMMN  | 0.8869 |        | 1.0000 | 1.0000 | 1.0000 | 1.0000 |
| Vine  | 0.0000 | 0.0000 |        | 0.1152 | 1.0000 | 1.0000 |
| Gauss | 0.0000 | 0.0000 | 0.8848 |        | 1.0000 | 1.0000 |
| Indep | 0.0000 | 0.0000 | 0.0000 | 0.0000 |        | 0.0000 |
| VAE   | 0.0000 | 0.0000 | 0.0000 | 0.0000 | 1.0000 |        |