# OpenReview forum: "Implicit Generative Copulas"
_NeurIPS.cc/2021/Conference — NeurIPS 2021 Poster_

### Official Review · Reviewer_qxq2 · 2021-07-11

**Rating:** 4
**Confidence:** 4

**Summary:**

The authors propose to model general copula models by training a neural network that samples from a latent space and maps samples to the high dimensional unit cube. Fitting the model is done by transforming the generated samples to have uniform margins by empirically computing the ranks. Then, the model is optimized by minimizing the MMD distance w.r.t true uniform data samples.

**Limitations And Societal Impact:**

Yes, he authors adequately addressed the limitations and potential negative societal impact of their work

**Main Review:**

Overall the paper is very well written and main results are clearly presented. However, my main concern is related to the applicability of  the proposed copula model. It seems that the proposed model can only be exploited as a generative model. For example, in order to compute CDFs which is the primary reason for learning copulas, the proposed model needs to generate samples, then empirically estimate the CDF which even with a large sample size could be not accurate in high dimensional settings.   Also,  computing the ranks seems to add errors to the training procedure which may affect  learning the model. On another front, the simulation results do not show substantial improvements over previous models (sometimes worse) especially Vine copula models. For the FashionMNIST dataset, it is probably more convenient to compare w.r.t the FID score as it is more used in image generation tasks.

Minor comments:
1. In line 36, that’s not accurate, there are also nested Archimedean copulas which support non-symmetry.
2. There is a repetition in line 76.
3. Typo in line 299.
4. Add standard deviation to the MMD scores.

**Time Spent Reviewing:**

6

---

> ### Author Response · Authors · 2021-08-10
> **Reply to Reviewer4 (qxq2 )**
>
> First of all, we would like to thank you for providing a thorough review.
> In the following we will first address your main concern of limited applicability of generative copula models and then comment on the other remarks.
>
> ### Limited Applicability
> The presented generative model is likelihood-free and you are right that this precludes various evaluations that could (efficiently) be performed with a tractable likelihood-based model.
> Nevertheless, in recent years, deep generative models that do not allow for an evaluation of the density have drawn significant attention in the range of modeling multivariate distributions (GANs, GMMNs, and VAEs).
> These publications include various application areas.
>
> We think that generative copula models could similarly be useful.
> They can be used in all the applications where generative models of multivariate distributions are used.
> They additionally offer the opportunity to a) better control the marginal distributions (by estimating them separately and possibly conditioning them on additional external factors) and b) to separately understand and control the dependency structure between the variables.
>
> Below we list some previous works and applications that employ copulas and that could benefit from our more versatile model.
> The CDF is not directly required in any of these.
>
> * **Forecasting and decision making under multivariate uncertainty**: Copulas are an established tool in forecasting applications to simulate from the predicted multivariate distribution, e.g. simulating returns of different asset (like the application on daily exchange rate returns presented in the paper) or simulating spatio-temporal weather scenarios [*1]. Samples from the predicted multivariate distribution can then be used as inputs for stochastic programming models, see e.g. [*2] for an application to the operation of a hydro power plant under uncertain market prices and water inflow.
> These methods are widely used in the financial and energy industry today and lead to direct financial benefits and efficiency gains.
> * **Multi-agent reinforcement learning**: Copulas can be used to simulate trajectories of multiple non-independent agents in reinforcement learning, see [*3].
> * **Synthetic tabular data generation**: Generative copula models are well suited for synthetic data generation tasks [*4]. Using synthetic data might result from the need to maintain data privacy or to emulate and enrich data obtained from physics based simulation models [*5].
> * **Image generation**: As presented in the paper, copulas can serve as a building block for generative image  models as in the Vine Copula Autoencoder [*6].
>
> We will take your critique as an indication to improve the introduction of the paper and motivate our approach more thoroughly.
>
> ### Computing the ranks:
> While using the ranks as an approximation of the CDFs is for sure introducing some error, it still provides an unbiased and consistent estimate of the current CDF that will become more accurate with increasing number of generated samples $M$.
> As $M$ can be treated as a hyper-parameter, we face a trade-off between accuracy of the CDF approximation and the training time.
> We could even chose a small $M$ at the beginning of the training and then fine-tune with an increasing sample size.
>
> ### Performance:
> We think that vines and TLL pair-copulas truly represent a very strong, state-of-the-art benchmark.
> So while we cannot in general beat the SOTA by a large margin and fall short to improve over it for the exchange rates data, we think the presented empirical results are promising as they show that our method is at least on-par with the SOTA and clearly improves over the often employed Gaussian copula as well as the approach proposed by [16] (see the updated results in our response to Reviewer2 (8HVZ)).
>
> ### Minor comments
>
> * **Nested Archimedean copulas:** You are right, nested Archimedean copulas are more flexible, we will change the formulation and be more precise here.
> * **Standard deviation for scores:** We added p-values for the scores of the FashionMNIST experiment based on the test proposed by Reviewer1. The differences in MMD scores are significant compared to all other models (see our reply to Reviewer1 (xntW)).
> We also report means and standard deviations for the XCR and MAGIC data from 5 repetitions over the 5 CV folds, please see our reply to Reviewer2 (8HVZ) for details.
>
> **References**
>
> [*1] A. Möller, A. Lenkoski, and T. L. Thorarinsdottir. Multivariate probabilistic forecasting using ensemble Bayesian model averaging and copulas. Quarterly Journal of the Royal Meteorological Society 139.673, 2013.
>
> [*2] S.E. Fleten and T.K. Kristoffersen. Short-term hydropower production planning by stochastic programming. Computers \& Operations Research 35.8: 2656-2671, 2008.
>
> [*3] H. Wang, L. Yu, Z. Cao, S. Ermon. Multi-Agent Imitation Learning with Copulas. In: European Conference on Machine Learning and Principles and Practice of Knowledge Discovery in Databases (ECML-PKDD), 2021.
>
> [*4] N. Patki, R. Wedge and K. Veeramachaneni. The Synthetic Data Vault. 2016 IEEE International Conference on Data Science and Advanced Analytics (DSAA): pp. 399-410, 2016.
>
> [*5] D. Meyer, T. Nagler, and R.J. Hogan. Copula-based synthetic data generation for machine learning emulators in weather and climate: application to a simple radiation model, Geosci. Model Dev. Discuss. [preprint], 2021.
>
> [*6] N. Tagasovska, D.Ackerer, and T. Vatter. Copulas as high-dimensional generative models: Vine copula autoencoders. In: Advances in Neural Information Processing Systems, 2019.

---

### Official Review · Reviewer_VacV · 2021-07-15

**Rating:** 7
**Confidence:** 5

**Summary:**

The paper proposes an implicit method (IGC) of estimating a copula. The suggested approach is elegant and from the presented results the method seems to work on par with existing non-parametric vine copula approaches. Previous works have made an attempt to use GANs for implicit copula estimation with difficulties, however, in IGC the authors suggest techniques to address the requirements for uniform margins with a simple two-step procedure leveraging the probability integral transform.
The general idea and contributions are clearly presented and I find this work valuable due to its simplicity and I believe its efficiency can be further exploited in other copula-related/inspired applications.

**Limitations And Societal Impact:**

The authors could mention the potential negative impact of generating fake data.

**Main Review:**

1. Since IGC is an implicit generative model, I would suggest comparing it to at least *one* such baseline (either [16] or [23] ). VAEs are considered Explicit generative models [1].

2. The latent dimension used in the experiments is quite low (up to 25 which suffices for fashionMNIST), did you try IGC for more complex datasets that require larger latent spaces? Was there any limitation?

3. The experiments could be extended to present this new method better. Toy examples such as Swiss Role or the multi-modal ring of Gaussians which are the difficult cases for implicit generative models. These are experiments easy to run and can be informative on the behavior of IGC (for example lack of mode collapse etc.)

4. The main advantage of copulas is that we can model the marginals and the dependence separately. Is this still possible with your approach?

5. Could you get some variance for the results in the tables by generating with different initializations?

6. How can IGC be extended to a conditional generative model?

[1] Goodfellow, Ian. "Nips 2016 tutorial: Generative adversarial networks." arXiv preprint arXiv:1701.00160 (2016).



------------------------------------------------------------
Update after Author Feedback and Discussion
------------------------------------------------------------

I thank the authors for the clarification. My questions and concerns have been mostly addressed/answered.
The original score still reflects my general opinion about the paper so I am keeping it, however, I do agree with the other reviews that it is important to show an advantage of the proposed method compared to previous copula-NN hybrids whether it is on new datasets or new applications/experiments.

**Time Spent Reviewing:**

8

---

> ### Author Response · Authors · 2021-08-10
> **Reply to Reviewer3 (VacV)**
>
> We would like to thank you very much for providing a thorough and useful review.
> We will answer to each of your points below.
>
> **1. Since IGC is an implicit generative model, I would suggest comparing it to at least one such baseline (either [16] or [23]). VAEs are considered Explicit generative models [1].**
>
> Your are right that VAEs are explicit models.
> Tagasovska et al. [38] already showed that the performance of the Vine Copula Autoencoder for image generation is comparable to DC-GANs and improves over standard VAEs in image generation.
> However, motivated by your's and the other reviewers' suggestions we re-ran our experiments on the XCR and MAGIC data sets and now also compare to [16], i.e. a generative model with a simple sigmoid output layer.
> In all cases the IGC model performs better than the model from [16].
> See our reply to Reviewer2(8HVZ) for details.
>
> **2. The latent dimension used in the experiments is quite low (up to 25 which suffices for fashionMNIST), did you try IGC for more complex datasets that require larger latent spaces? Was there any limitation?**
>
> We so far did not perform experiments in dimensions greater than 25. However, we do not see any direct limitations for higher dimensions that would not apply also for GANs or GMMNs that have been shown to work well in higher dimensions, given sufficient training data.
>
> **3. The experiments could be extended to present this new method better. Toy examples such as Swiss Role or the multi-modal ring of Gaussians which are the difficult cases for implicit generative models. These are experiments easy to run and can be informative on the behavior of IGC (for example lack of mode collapse etc.)**
>
>  This is a good point. We indeed did some experiments on toy data sets like these in the very beginning. We decided to focus on synthetic experiments with copulas because we think that it is important to show that the IGC approach is SOTA compared with other methods to explicitly model copulas as these methods are the direct competitors. However, we agree that providing these standard experiments will improve the presentation and we will add them in the appendix of the final version.
>
> **4. The main advantage of copulas is that we can model the marginals and the dependence separately. Is this still possible with your approach?**
>
> Yes! Given a data set we can first obtain the "pseudo-observations" in the unit space by estimating the marginal CDFs of the data and transforming component-wise.
> This can be done independently of the training of the IGC model.
> However, as our model is differentiable, we could also learn marginals and copula simultaneously in an end-to-end fashion.
>
> **5. Could you get some variance for the results in the tables by generating with different initializations?**
>
> In the updated experiments we report means and standard deviations across 5 different initializations and the 5 CV folds (see our reply to Reviewer2 (8HVZ)).
> For FashionMNIST we employ the test statistic suggest by Reviewer1 to assess statistical differences in the MMD scores (see our reply to Reviewer1 (xntW) for details).
>
> **6. How can IGC be extended to a conditional generative model?**
>
> We see two straightforward ways: (a) concatenate the exogenous inputs to the noise inputs or (b) condition the parameters of the noise distribution on the exogenous inputs.

---

### Official Review · Reviewer_8HVZ · 2021-07-15

**Rating:** 7
**Confidence:** 5

**Summary:**

The paper proposes a novel method to model "high"-dimensional copulas using deep neural networks.
They are able to train their implicit generative models in a likelihood-free way and obtain performances that are on-par with SOTA approaches.

**Limitations And Societal Impact:**

The authors have not mentioned anything.

**Main Review:**

Given the noted lack of availability of performant (>2)-dimensional copula models, I welcome this contribution.
I think that it is a worthwhile pursuit and that the authors' approach is novel and interesting.
Additionally, the paper is well written and reads nicely.

One thing that bothers me is that there is no theoretical result at all.
No universal approximation/sample complexity/generalization bound on the ML side, no identifiability/consistency/asymptotica distribution/estimation rate on the stats side.

Another concern that I have is that I see no way of obtaining computing copula densities or probabilities in closed form, but this is true for any implicit generative model.
Similarly, I don't see a way to either marginalize or condition over subsets of the variables, but maybe the authors can expand on this?

Below is my section per section review.

## Introduction

The introduction is good, and the literature review cites all the important papers on the topic.
However, the "contributions" part need serious work.

### Uniformity property

Why is ensuring the uniformity property in finite samples important?
As a copula expert/statistician foremost and ML researcher secondly, I think that for a copula model to be sensible, being approximately uniform in finite sample is enough as long as it is asymptotically so.
This is the case for most nonparametric copula estimators such as the bivariate kernel density estimation used in [31].
See e.g., [12], which powers the TLL method and can easily be generalized to higher-dimensions.

Note that, if D is a function that satisfies all the requirements of a copula except the uniform margins, i.e. if $D:[0,1]^n\\to[0,1]$ is s.t. (1) $D(u_1, \\dots, u_{i-1}, 0, u\_{i+1}, \\dots, u_n) = 0$, (2) $D(1, \\dots, 1) = 1$, and (3) D is n-increasing, then $F(x_1, \\dots, x_n) = D(F_1(x_1), \\dots, F_n(x_n))$ is a valid distribution with margins $F_1, \\dots, F_n even$ if $D(1, \\dots, 1, u_i, 1, \\dots, 1) \\neq u_i$.

Therefore, since ensuring marginal uniformity is the "key challenge", its importance should be motivated.
It seems that ML people designing new copula models, whether in a generative fashion or via deep nets, focus a lot on this without really thinking about how relevant it is (if at all).
So why is this important and/or relevant?
And if it is not, then what is this contribution really bringing as an improvement over [16].

As a side-note: since you are using the softrank layer to approximate the ECDF of the latent variables during training, the model that you really learn is only an approximation of the copula.
And estimating the latent marginals after training is an example of a posteriori normalization that could be applied to other methods (e.g., [16]) as well.

### No tree structures

Why is the fact that the model is free from tree structures relevant as you mention in the main contributions?
In the nonparametric context, trees represent a structural constraint allowing to obtain better estimation rates (e.g., bivariate, see "Evading the curse of dimensionality in nonparametric density estimation with simplified vine copulas").
In your case, since there is no structural constraint, your model will suffer from the curse of dimensionality in the sense that you will need a sample size that grows exponentially with the dimension to obtain the same error rates.

### Misleading claim of a simpler framework

In my opinion, the claim of "similar or improved performance for different tasks, with a structurally simpler framework" is at best misleading.
I am not saying that vine structures are "simple", but how is a deep neural network "simpler" than a nested set of trees.
Also, there is well established software for vines that is ready to use, and the following seems easy enough to me:

```
import pyvinecopulib as pv
fit = pv.Vinecop(data)
```

And training times for vines are order of magnitudes smaller than the ones for your method.
Thus, I would argue to either clarify what is "simpler" about your approach or tamp down the claim.

### Likelihood-free training

Please explain and motivate why this is a good thing.
What you do is essentially minimize a distance between your copula and the empirical copula.
But it is well known that MLEs are efficient (i.e., attaining the Cramer-Rao Lower Bound), so likelihood-based training is generally viewed as desirable if at all possible.
To me, it seems that the reason you are training likelihood-free is simply that you cannot obtain the density.

## Proposed IGC Model

### Copula basics

- Why write $F_{X,d}$ instead of $F_{X_d}$?
- Please mention whether you are concerned with discrete or continuous data (Sklar's theorem also giving unicity of the copula for continuous distributions).

### ICG model

- Please describe the architecture that you use in more details. This could also be done in the next section.

## Training

### Parameter estimation

- Have you tried optimal transport-based distances (e.g., Wassertein)? It would seem like the natural thing to do in your context.

## Experiments

### Synthetic data

- 1000 observations for a bivariate dataset is a fairly large sample already. It would be interesting to see how the methods improve from low to high samples (e.g., 100, 200, 500, 1000, 2000, 5000),
- In the Gaussian mixture example, the only reason ICG does better than TLL is because the automatic bandwidth selection rule of the later is geared towards unimodal distributions. Also, the TLL method is generally more stable than TLL1/TLL2 for complex (e.g., multimodal) distributions.
- Regarding your discussion about recovering the true model and ICG doing better than vines when the true model is a vine: the reason is not related to the likelihood/misspecification of the pair-copulas. It is due to the fact that the selected tree structure is incorrect (you might know that the space of vine structures is super exponential, e.g. much larger than the space of DAG structures). Basically, the Dissmann algorithm is a greedy algorithms whose heuristic aims at capturing most of the dependence in the higher trees. But your simulation setup creates models by sampling parameters irrespectively of the tree level. Thus, the structure selection algorithm is simply not designed for the model class that you sample from. I'm not saying it's bad, just that it is unsuprising. This also likely explains why the Vine(TLL2) is OK most of the time (i.e., low median ISE), but can go really wrong is a poor structure is selected (i.e., doesn't matter how well you estimate pairs if the structure is bad).

### Real data

- Table 1 is missing the standard errors. In particular, for MAGIC, it seems like Vine, Vine(TLL2) and IGC differences are unlikely to matter. Given that you do cross-validation, adding the standard errors should be easy enough.

- Table 2, same comment, but here you say "significantly"...

**Time Spent Reviewing:**

3

---

> ### Author Response · Authors · 2021-08-10
> **Reply to Reviewer2 (8HVZ)**
>
> We would like to express our gratitude for your detailed and thoughtful review and the many useful suggestions you have made.
>
> ## Main review
> * **No theoretical results**: At the end of section 2 we argue for universal approximation properties of our approach. We did not highlight it as a key point of our paper, since it follows quite directly from the universality of deep generative models [26]. We will make it more visible in the final version of the paper.
>
> * **Obtaining densities, marginalization/conditioning on groups of variables**: As many generative approaches (GANs, GMMNs, VAEs) our approach is limited in this respect. As discussed in the reply to Reviewer4 (qxq2) there are, however, many practical and important applications where being able to sample is sufficient. We will also make this point clearer in the final version.
>
> ## Introduction
>
> ### Uniformity property:
> Thank you a lot for the detailed discussion on this matter!
>
> We started our research without the softrank layer but instead used an approach almost identical to [16], i.e. we simply used a sigmoid output layer to map to $[0,1]^D$.
> We then found that the learned distributions often had marginal distributions rather far from uniformity.
> We see this as serious problem since an important application of copula models is to be able to control the marginals in the data space and the dependency structure separately.
> If the copula model is far from marginal uniformity, the marginal distributions in data space are not as desired and this can be a problem for many applications (e.g. in financial trading we think that risk control of an individual asset is mandatory before studying correlation effects).
> We will motivate this point more thoroughly in the introduction.
>
> Furthermore, the softrank layer allows the model to have arbitrary latent marginals, i.e. it can choose marginal distributions that are "convenient" to learn the desired dependency structure.
> In the case of a sigmoid output layer we require from the model to approximate a $Logistic(0,1)$ for all marginals to ensure a uniform distribution after applying the sigmoid.
>
> To further illustrate the effectiveness of the approach, we have have rerun the experiments for XCR and MAGIC, now also comparing our approach to [16], i.e. the softrank layer is replaced with a simple sigmoid output layer.
>
> Here are the results:
>
> | | XCR | MAGIC |
> | ------------- | ------------- | ------------- |
> | GAUSS | $0.9196(\pm 0.1024)$ | $0.2415 (\pm0.01672)$ |
> | VINE | $0.2457(\pm 0.0383)$ | $0.0588 (\pm0.00550)$ |
> | [16] | $0.5791 (\pm 0.2638) $| $0.08065 (\pm0.02863)$ |
> | IGC(ours) | $0.3829 (\pm0.1385)$ | $0.03448(\pm0.01051)$ |
>
> (ISE mean and standard deviation from 5 folds and 5 initializations per fold, i.e. we average 25 results for the NN models and 5 results for the other approaches, we used Scikit learn CV random seed=42, all neural networks use 100 neurons, 2 layers, 500 epochs (Adam), ReLU activation, i.e. the same architecture as in the synthetic experiments.)
>
> The IGC approach still comes out on top for MAGIC and shows better performance than [16] for both data sets.
> Additionally the standard deviations are also smaller.
>
> ### No tree structures:
> As you mention, the tree search space quickly becomes very large and a wrong choice can lead to bad performance.
> This is why we argue that avoiding trees might be beneficial.
> Of course, if the problem setting already dictates a certain tree structure, this argument does not hold.
> We will try to make this claim more precise in the paper.
> Moreover, tree search is not a differentiable operation and hence prohibits end-to-end learning, e.g. in applications in reinforcement learning.
>
> ### Simpler framework:
> This might be a matter of perspective.
> Seen from a machine learning perspective, training a "vanilla" deep NN with an additional soft-rank layer is easy to understand and fits nicely with the commonly used tools.
> We will rephrase this claim more carefully.
> Software-wise, both approaches can be applied with very compact code.
>
> ### Likelihood free training:
> We agree that MLE based training has a lot of advantages.
> Still, training with MMD/Energy distance provides a statistically sound objective and our approach extends the available model class for copulas at the expense of not being able to train with MLE.
>
> ## Proposed model:
> * We will add that we are concerned with continuous data
> * **Architecture**: We describe the architecture we used for the experiments in section 4.
>
> ## Training:
> * We only experimented with MMD type losses.
>
> ## Experiments:
> * **Samples sizes for bi-variate data**: This is a good point. We will add results for different training sample sizes.
> * **Gaussian mixture/vine structure**: Thank your very much for providing these insights, we will adapt the discussion for the Gaussian mixture copula and the vine models to accommodate you comments

---

### Official Review · Reviewer_xntW · 2021-07-16

**Rating:** 6
**Confidence:** 2

**Summary:**

The paper proposes a novel method for learning flexible cupola distributions of high dimensional data using implicit generative models, the first non-parametric method for estimating cupola distributions with a guarantee of uniform marginal distributions.


**Limitations And Societal Impact:**

The authors discuss some limitations related to computational scaling.

**Main Review:**

The paper is generally well written and I found the description of the model and training procedures easy to follow. The authors provide a thorough review of related literature and put their work into context.

I am not closely familiar with the literature on estimating cupolas but based on the authors discussion the method seems novel and mathematically sound. The main contribution of the work are ensuring that the cupola distributions learned by the implicit generative model has uniform marginals by using a differentiable approximation of the cumulative density function (a softrank layer) to transform the output of the generator network into $[0,1]^D$.

The method is evaluated on a number of synthetic and real-world datasets and shown to be matching or in some cases out-performing SOTA methods such as vine cupolas.
The weaknesses of the evaluation is that it relies on estimating CDF functions in up to 5-dimensional spaces which can be inaccurate. For fashion MNIST the authors report MMD distances as point estimates with relatively small differences between methods. It would be useful to get estimates of variability either by resampling or performing a 3-sample test as described by Bounliphone et al 2015 (A Test of Relative Similarity For Model Selection in Generative Models.)


**Time Spent Reviewing:**

2.5h

---

> ### Author Response · Authors · 2021-08-10
> **Reply to Reviewer1 (xntW)**
>
> Thank your very much for a thorough and constructive review.
>
> We agree that using the empirical CDFs introduces some variability in the evaluation. However, we found that using $10^5$ samples to estimate the ISE leads to stable results.
> Furthermore, we updated our experiments for the exchange rates and magic data sets and now additionally report means and standard deviations over the 5 CV-folds and 5 different random weight initializations for the NNs.
> Qualitatively the rankings of the methods did not change.
> The results of this update are discussed in detail in the response to Reviewer2 (8HVZ).
>
> Additionally, we followed your suggestion and computed p-values for the MMD scores for the FashionMNIST experiment via the test proposed by Bounliphone et al., here are the results:
>
> * image space (IGC-VINE: $<10^{-6}$, IGC-GAUSS:$<10^{-6}$, IGC-INDEP:$<10^{-6}$, IGC-VAE: $<10^{-6}$)
> * latent space (IGC-VINE: $0.0188$, IGC-GAUSS:$<10^{-6}$, IGC-INDEP:$<10^{-6}$)
> * copula space (IGC-VINE: $0.0129$, IGC-GAUSS: $0.0010$, IGC-INDEP:$<10^{-6}$)
>
> We will add the p-values for all comparisons to the final version.

---

### Decision · Program_Chairs · 2021-09-27

**Decision:**

Accept (Poster)

**Comment:**

Reviewers had divergent opinions on this paper — behind the scenes, in the discussion phase, one was arguing strongly for acceptance, and one for rejection. The primary complaints were: (i) the model may have limited applicability, due to the fact that it is purely "implicit" and does not provide a density estimate, precluding many of the common uses of copula models; (ii) the empirical evaluation could have been stronger, as there is only moderate improvement relative to vine copula baselines, and for the image data such as fashion MNIST other metrics could be more appropriate. However, all reviewers agreed that the authors responded well to concerns during the discussion period. I would strongly suggest including the suggested additional results on generation for standard datasets in the appendix, as mentioned, as well as including more on motivating examples in which an implicit copula model is helpful despite lack of an explicit density.